OBSERVATION

# Antibody Response to the BA.5 Bivalent Vaccine Shot: a Two-Year Follow-Up Study following Initial COVID-19 mRNA Vaccination

Yosuke Hirotsu,[a] Hiroki Sugiura,[b] Mika Takatori,[c] Hitoshi Mochizuki,[a,d,e] Masao Omata[e,f]

[a]Genome Analysis Center, Yamanashi Central Hospital, Kofu, Japan
[b]Division of Clinical Biochemistry and Immunology, Yamanashi Central Hospital, Kofu, Japan
[c]Division of Infection Control and Prevention, Yamanashi Central Hospital, Kofu, Japan
[d]Central Clinical Laboratory, Yamanashi Central Hospital, Kofu, Japan
[e]Department of Gastroenterology, Yamanashi Central Hospital, Kofu, Japan
[f]The University of Tokyo, Tokyo, Japan

**ABSTRACT** Although many studies have been conducted on the increase in spike antibody levels after vaccination, there is insufficient prospective and longitudinal information on the BA.5-adapted bivalent vaccine up to the fifth vaccination. In this study, we conducted a follow-up study of spike antibody levels and infection history in 46 health care workers who received up to 5 vaccinations. Monovalent vaccines were administered for the first to fourth vaccinations, and a bivalent vaccine was administered for the fifth vaccination. 11 serum samples were collected from each participant, and antibody levels were measured in a total of 506 serum samples. During the observation period, 43 of the 46 health care workers had no infection history, and 3 had a history of infection. Spike antibody levels peaked at 1 week after the second booster vaccination and gradually declined until the 27th week after the second vaccination. After 2 weeks following the fifth BA.5-adapted bivalent vaccine, the spike antibody levels significantly increased (median: 23,756 [IQR: 16,450 to 37,326]), compared to those measured before vaccination (median: 9,354 [IQR: 5,904 to 15,784]) (paired Wilcoxon signed-rank test, $P = 5.7 \times 10^{-14}$). These changes in antibody kinetics were observed regardless of age or sex. These results suggest that booster vaccination increased the spike antibody levels. Regular vaccination is effective in maintaining long-term antibody levels.

**IMPORTANCE** A COVID-19 bivalent mRNA vaccine was developed and administered to health care workers. The COVID-19 mRNA vaccine induces a robust antibody response. However, little is known about the antibody response to vaccines in serially collected blood samples from the same individuals. Here, we provide two-year follow-up data on the humoral immune response to COVID-19 mRNA vaccines in health care workers who received up to five vaccinations, including the BA.5-adapted bivalent vaccine. The results suggest that regular vaccination is effective in maintaining long-term antibody levels and have implications for vaccine efficacy and booster dose strategies in health care settings.

**KEYWORDS** spike, bivalent vaccine, antibody, mRNA vaccine, COVID-19

Vaccination is a key approach by which to control COVID-19 (1–3). Antibodies from vaccination protect against infection, but new variants, such as Omicron, evade antibodies (4–6), which leads to reinfection (7). Bivalent mRNA vaccines for Omicron and the ancestral strain boost immunity (8–13), but it is unclear how long their effects last for health care workers (HCWs). We previously reported short-term results of the COVID-19 mRNA vaccine for 7 weeks after the first vaccination (4 weeks after the second booster shot) (14). Our follow-up study investigates the vaccine-induced immune response durability in HCWs. We

Address correspondence to Yosuke Hirotsu, hirotsu-bdyu@ych.pref.yamanashi.jp.

The authors declare no conflict of interest.

**TABLE 1** Patient characteristics and spike antibody levels

| Characteristic | $n = 46^a$ |
|---|---|
| Age, median (IQR) | 48 (33 to 60) |
| | |
| Age class, *n* (%) | |
| 20s | 6 (13%) |
| 30s | 10 (22%) |
| 40s | 8 (17%) |
| 50s | 9 (20%) |
| 60s | 13 (28%) |
| | |
| Sex, *n* (%) | |
| Female | 14 (30%) |
| Male | 32 (70%) |
| | |
| V1 | |
| Pfizer-BioNTech, n (%) | 46 (100%) |
| | |
| V2 | |
| Pfizer-BioNTech, n (%) | 46 (100%) |
| | |
| V3 | |
| Pfizer-BioNTech, n (%) | 46 (100%) |
| | |
| V4 | |
| Moderna | 31 (67%) |
| Pfizer-BioNTech, n (%) | 15 (33%) |
| | |
| V5 | |
| Pfizer-BioNTech, n (%) | 46 (100%) |
| | |
| Anti-S (U/mL), median (IQR) | |
| 3W | 51 (21 to 118) |
| 4W | 1,959 (860 to 2,875) |
| 5W | 1,932 (1,089 to 2,381) |
| 6W | 1,431 (844 to 1,963) |
| 7W | 1,179 (712 to 1,648) |
| 12W | 812 (471 to 1,135) |
| 30W | 535 (310 to 726) |
| 52W | 12,178 (8,160 to 20,191) |
| 100W | 9,354 (5,904 to 15,784) |
| 102W | 23,756 (16,450 to 37,326) |

[a]IQR, interquartile range; V1, first vaccination; V2, second vaccination; V3, third vaccination; V4, fourth vaccination; V5, fifth vaccination; U, unit.

report on the humoral immune response over a 102-week period in 46 healthy HCWs, shedding light on antibody kinetics and vaccine efficacy for booster doses.

We studied 46 HCWs (median age: 48 [IQR: 33–60]; 14 females and 32 males) who received 5 vaccine doses (Table 1). The age groups were as follows: 20s (*n* = 6), 30s (*n* = 10), 40s (*n* = 8), 50s (*n* = 9), and 60s (*n* = 13). Among these 46 individuals, 3 HCWs had a history of SARS-CoV-2 infection until the fifth vaccination, and 43 HCWs were naive individuals with no history of infection. We collected serum samples from all participants for approximately 2 years, and we measured their spike antibody levels to examine the longitudinal humoral immune response after the first vaccine dose (Fig. S1; Table S1). The median spike antibody levels after vaccination were 51 (IQR: 21 to 118) at 3 wk, 1,959 (IQR: 860 to 2,875) at 4 wk, 1,932 (IQR: 1,089 to 2,381) at 5 wk, 1,431 (IQR: 844 to 1,963) at 6 wk, 1,179 (IQR: 712 to 1,648) at 7 wk, 812 (IQR: 471 to 1,135) at 12 wk, 535 (IQR: 310 to 726) at 30 wk, 12,178 (IQR: 8,160 to 20,191) at 52 wk, 9,354 (IQR: 5,904 to 15,784) at 100 wk, and 23,756 (IQR: 16,450 to 37,326) at 102 wk (Table 1).

The trend of the spike antibody levels showed a peak at 4 wk (1 week after V2), and this was followed by a decrease until 30 wk (27 weeks after V2). The levels increased again at 52 wk (11 weeks after V3) and showed a further increase at 102 wk (2 weeks after V5$_{bivalent}$) (Fig. 1A). This trend was observed regardless of age and sex (Fig. 1B and C).

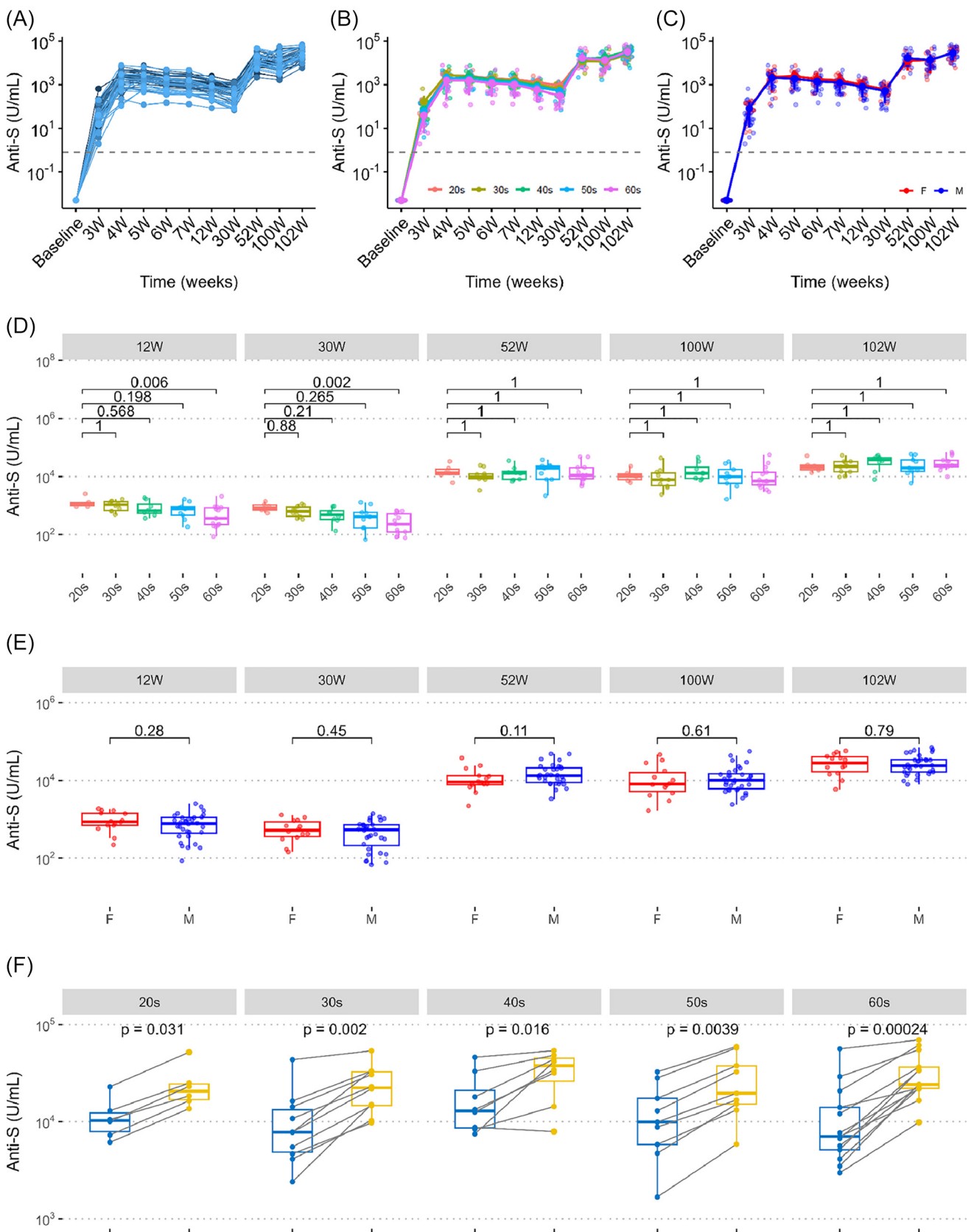

**FIG 1** Trend of anti-spike antibodies after a booster vaccination. Serial changes in the serum anti-spike (S) antibody levels (U/mL) from the first vaccine administration. (A and B) The levels were evaluated at 11 time points: baseline, 3 wk, 4 wk, 5 wk, 6 wk, 7 wk, 12 wk, 30 wk, 52 wk, 100 wk, and 102 wk.

The monovalent mRNA vaccine for V4 included both the Moderna ($n = 31$) and Pfizer-BioNTech ($n = 15$) vaccines, and there was no significant difference in the overall trend of the spike antibody levels between the two vaccines (Fig. S2A). There was also no significant difference in the spike antibody levels at 100 wk and 102 wk (Fig. S2B) ($P = 0.15$ at 100 weeks, $P = 0.91$ at 102 weeks, Wilcoxon rank-sum test).

We previously reported data on the age and sex differences between individuals from 1 wk to 7 wk after the COVID-19 mRNA vaccine (14). In this study, we analyzed the age and sex differences between the spike protein levels in serum collected at later time points. At 12 wk (9 weeks after V2) and 30 wk (27 weeks after V2), the spike antibody levels were significantly lower in HCWs in their 60s, compared to those in their 20s (Fig. 1D) (Bonferroni-adjusted Wilcoxon rank-sum test, 12 wk, $P_{adj} = 0.006$; 30 wk, $P_{adj} = 0.002$). However, at 52 wk (11 weeks after V3), 100 wk (immediately before $V5_{bivalent}$), and 102 wk (2 weeks after $V5_{bivalent}$), there were no significant differences in the spike antibody levels between any age group, compared to those in their 20s (Fig. 1D) (Bonferroni-adjusted Wilcoxon rank-sum test, $P_{adj} = 1$). Additionally, there was no significant difference in the spike antibody levels between males and females at 12 wk, 30 wk, 52 wk, 100 wk, and 102 wk (Fig. 1E) (Wilcoxon rank sum test, 12 wk, $P = 0.28$; 30 wk, $P = 0.45$; 52 wk, $P = 0.11$; 100 wk, $P = 0.61$; 102 wk, $P = 0.79$).

All HCWs received the BA.5-adapted bivalent vaccine (Pfizer-BioNTech) for the fifth vaccination. Changes in the spike antibody levels were analyzed before and after $V5_{bivalent}$. Among all 46 participants, except for 1, the spike antibody levels increased after vaccination, compared to before vaccination. A significant increase in the spike antibody levels was observed at 102 wk (2 weeks after $V5_{bivalent}$), compared to 100 wk (just before $V5_{bivalent}$) (Fig. S3A) (paired Wilcoxon signed-rank test, $P = 5.7 \times 10^{-14}$), indicating a robust immune response to the bivalent vaccine. The median difference in S antibody levels between 100 wk and 102 wk was 13,624 (IQR: 8,344 to 21,403). When evaluated by age group, the spike antibody levels increased after the BA.5-adopted bivalent vaccine in all age groups (Fig. 1F) (paired Wilcoxon signed-rank test, 20s, $P = 0.031$; 30s, $P = 0.002$; 40s, $P = 0.016$; 50s, $P = 0.0039$; 60s, $P = 0.00024$). Similarly, there were significant increases in the antibody levels in both males and females (Fig. S3B) (paired Wilcoxon signed-rank test, females, $P = 0.00012$; males, $P = 9.3 \times 10^{-10}$).

Our findings indicate that antibody levels decrease over time after vaccination (15–18). Furthermore, individuals in their 60s had significantly lower levels at 12 wk and 30 wk after the booster vaccination, which is consistent with the results of previous reports on the duration of the immune response in the elderly (19, 20). However, at 52 wk and 100 wk, there was no significant difference in the spike antibody levels between age groups, indicating that regular vaccinations help maintain sufficient antibody levels. The BA.5-adapted bivalent vaccine provided a booster effect in all age groups. Our study highlights the importance of monitoring antibody levels over time and the potential benefits of booster vaccinations, especially for high-risk individuals. Additional booster shots may be necessary, particularly for older adults who experienced a decline in antibody levels at some point (21).

**Data availability.** The source data for the figures and tables are provided in Table S2.

**FIG 1** Legend (Continued)
The line plots show the (A) data from all 46 HCWs as well as the mean and standard deviation of the spike antibody levels for (B) the different age groups (20s [$n = 6$], 30s [$n = 10$], 40s [$n = 8$], 50s [$n = 9$], and 60s [$n = 13$]) and for (C) females ($n = 14$) and males ($n = 32$). Each dot represents the antibody level of an individual HCW. The dashed line represents the cutoff value of the spike antibody titer at 0.8 U/mL. (D and E) Box plots show the differences in the serum anti-S antibody levels (U/mL) at each time point from 12 wk to 102 wk, stratified by (D) age group (20s, 30s, 40s, 50s, and 60s) and (E) sex. The data by age group were obtained using the 20s group as the reference. The statistical analyses were performed using Wilcoxon rank-sum tests, and the $P$ values were adjusted with the Bonferroni correction for multiple testing. F, female; M, male. (F) The increase in the S antibody levels in HCWs who were aged from their 20s to their 60s and received the BA.5 adapted bivalent vaccine. Box plots show the data before (100 wk) and 2 weeks after (102 wk) the administration of the BA.5 adapted bivalent vaccine. The statistical analyses were performed using paired Wilcoxon signed-rank tests. Each box indicates the interquartile range (top, the third quartile; bottom, the first quartile), with a horizontal line indicating the median.

## SUPPLEMENTAL MATERIAL

Supplemental material is available online only.

**SUPPLEMENTAL FILE 1**, PDF file, 0.33 MB.
**SUPPLEMENTAL FILE 2**, XLSX file, 0.01 MB.

## ACKNOWLEDGMENTS

We thank the clinical laboratory technicians in the microbiology laboratories at our institution. This study was supported by a Grant-in-Aid for the Genome Research Project from Yamanashi Prefecture (to M.O. and Y.H.), the Japan Society for the Promotion of Science (JSPS) KAKENHI Early-Career Scientists JP18K16292 (to Y.H.), a Grant-in-Aid for Scientific Research (B) 20H03668 and 23H02955 (to Y.H.), a Research Grant for Young Scholars (to Y.H.), the YASUDA Medical Foundation (to Y.H.), the Uehara Memorial Foundation (to Y.H.), a Medical Research Grant from the Takeda Science Foundation (to Y.H.), and the Kato Memorial Bioscience Foundation (to Y.H.).

Y.H. drafted the manuscript, visualized the data, and performed the statistical analyses. H.S. measured the antibody levels and collected the data. M.T. collected the data from the HCW, including vaccination status and infection history. H.M. organized and supervised the study. M.O. conceptualized the study design and revised the manuscript. All authors reviewed and approved the manuscript.

We declare no competing interests.

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
