## [Reviewer comments · Microbiology Spectrum]

Microbiology Spectrum

Antibody response to the BA.5 bivalent vaccine shot: a two-year follow-up study following initial COVID-19 mRNA vaccination

Yosuke Hirotsu, Hiroki Sugiura, Mika Takatori, Hitoshi Mochizuki, and Masao Omata

Corresponding Author(s): Yosuke Hirotsu, Yamanashi Central Hospital

Review Timeline:

Submission Date:	March 27, 2023
Editorial Decision:	April 24, 2023
Revision Received:	April 25, 2023
Editorial Decision:	April 26, 2023
Revision Received:	April 27, 2023
Accepted:	April 27, 2023

Editor: Takamasa Ueno

Reviewer(s): Disclosure of reviewer identity is with reference to reviewer comments included in decision letter(s). The following individuals involved in review of your submission have agreed to reveal their identity: Marco Becherelli (Reviewer #2)

Transaction Report:

DOI: <https://doi.org/10.1128/spectrum.01316-23>

April 24, 2023

Dr. Yosuke Hirotsu
Yamanashi Central Hospital
Genome Analysis Center
1-1-1 Fujimi
Kofu, Yamanashi
Japan

Re: Spectrum01316-23 (Antibody response to the BA.5 bivalent vaccine shot: a two-year follow-up study following initial COVID-19 mRNA vaccination)

Dear Dr. Yosuke Hirotsu:

Thank you for submitting your manuscript to Microbiology Spectrum. As you will see your paper is very close to acceptance. However, because of the fine-focused scope of this study, I would like the authors to resubmit the revised manuscript to 'Observations' category, instead of 'Research Articles'. Please modify the manuscript along the lines I have recommended. As these revisions are quite minor, I expect that you should be able to turn in the revised paper in less than 30 days, if not sooner. If your manuscript was reviewed, you will find the reviewers' comments below.

When submitting the revised version of your paper, please provide (1) point-by-point responses to the issues raised by the reviewers as file type "Response to Reviewers," not in your cover letter, and (2) a PDF file that indicates the changes from the original submission (by highlighting or underlining the changes) as file type "Marked Up Manuscript - For Review Only". Please use this link to submit your revised manuscript. Detailed instructions on submitting your revised paper are below.

Link Not Available

Sincerely,

Takamasa Ueno

Reviewer comments:

Reviewer #2 (Comments for the Author):

The manuscript "Antibody response to the BA.5 bivalent vaccine shot: a two-year follow-up study following initial COVID-19 mRNA vaccination" examined the spike antibody level over a period of two years in healthcare workers receiving 5 doses of COVID-19 mRNA vaccine. The manuscript shows that booster vaccination is effective in maintaining long term Spike antibody level without difference in any age group. In addition, a significant increase in Spike antibody level was observed after the 5th dose with the bivalent vaccine. Overall, the manuscript is well written, and the data showed are consistent with the scope of the study.

Comments:

- It has been reported that after the 5th vaccination there is a significant increase in Spike antibody level but the antibody titer after the 4th vaccination is only evaluated before the 5th vaccination. Do the authors have data on the trend titer decline between the 4th and 5th vaccination?
- It would be interesting to evaluate following the 5th vaccination with a bivalent vaccine how the neutralizing antibodies against the different variants change over the time.
- I would suggest including a note in table S1 or in the text regarding the timing of the 4th vaccination.

Preparing Revision Guidelines

Please return the manuscript within 60 days; if you cannot complete the modification within this time period, please contact me. If you do not wish to modify the manuscript and prefer to submit it to another journal, please notify me of your decision immediately so that the manuscript may be formally withdrawn from consideration by Microbiology Spectrum.

The manuscript “Antibody response to the BA.5 bivalent vaccine shot: a two-year follow-up study following initial COVID-19 mRNA vaccination” examined the spike antibody level over a period of two years in healthcare workers receiving 5 doses of COVID-19 mRNA vaccine. The manuscript shows that booster vaccination is effective in maintaining long term Spike antibody level without difference in any age group. In addition, a significant increase in Spike antibody level was observed after the 5th dose with the bivalent vaccine. Overall, the manuscript is well written, and the data showed are consistent with the scope of the study.

Comments:

- It has been reported that after the 5th vaccination there is a significant increase in Spike antibody level but the antibody titer after the 4th vaccination is only evaluated before the 5th vaccination. Do the authors have data on the trend titer decline between the 4th and 5th vaccination?
- It would be interesting to evaluate following the 5th vaccination with a bivalent vaccine how the neutralizing antibodies against the different variants change over the time.
- I would suggest including a note in table S1 or in the text regarding the timing of the 4th vaccination.

Responses to the comments of Reviewer #2

【Comment】

The manuscript "Antibody response to the BA.5 bivalent vaccine shot: a two-year follow-up study following initial COVID-19 mRNA vaccination" examined the spike antibody level over a period of two years in healthcare workers receiving 5 doses of COVID-19 mRNA vaccine. The manuscript shows that booster vaccination is effective in maintaining long term Spike antibody level without difference in any age group. In addition, a significant increase in Spike antibody level was observed after the 5th dose with the bivalent vaccine. Overall, the manuscript is well written, and the data showed are consistent with the scope of the study.

【Response】

Thank you for your peer review. We appreciate your constructive comments. We have revised the manuscript according to your suggestions.

【Comment】

- It has been reported that after the 5th vaccination there is a significant increase in Spike antibody level but the antibody titer after the 4th vaccination is only evaluated before the 5th vaccination. Do the authors have data on the trend titer decline between the 4th and 5th vaccination?

【Response】

We don't have data on the decline trend between the 4th and 5th vaccinations because our experimental design was incomplete. The 4th vaccination was administered between 66-76 weeks after the 1st vaccination. If we collected the sera from healthcare workers and got the data after the 4th vaccination (around 70-80 weeks), we would answer your comments, but we cannot present data unfortunately. Perhaps the spike antibody titer was upregulated after the 4th vaccination and is expected to decrease from after the 4th vaccination until just before the 5th vaccination.

【Comment】

- It would be interesting to evaluate following the 5th vaccination with a bivalent vaccine how the neutralizing antibodies against the different variants change over the time.

【Response】

We completely agree with your comment. It is a very interesting point to evaluate the neutralizing antibody against different variants before and after the 5th vaccination. We don't have data on neutralizing antibodies against variants. However, recent studies showed that several types of emerging Omicron BQ.1.1 and XBB.1.5 variants are resistant to bivalent vaccine (Lasrado et al., bioRxiv. 2023, doi: 10.1101/2023.01.22.525079.; Jing Zou et al.,

NEJM. 2023, 388(9):854-857; Qu et al., Cell Reports. 2023, doi: <https://doi.org/10.1016/j.celrep.2023.112443>). According to these previous reports, bivalent vaccine has become less effective against some circulating variant with acquiring new mutation in Spike protein.

【Comment】

- I would suggest including a note in table S1 or in the text regarding the timing of the 4th vaccination.

【Response】

The 4th vaccination was administered between 66-76 weeks after the 1st vaccination. We noted the timing of 4th vaccination in Table S1.

April 26, 2023

Dr. Yosuke Hirotsu
Yamanashi Central Hospital
Genome Analysis Center
1-1-1 Fujimi
Kofu, Yamanashi
Japan

Re: Spectrum01316-23R1 (Antibody response to the BA.5 bivalent vaccine shot: a two-year follow-up study following initial COVID-19 mRNA vaccination)

Dear Dr. Yosuke Hirotsu:

Thank you for resubmitting the revised version. I think that the revised version well addressed the comments of the reviewer and the scientific evaluation has been well done. However, Observation category requires the guideline '1,200 words with a maximum of 2 figures/tables and 25 references'. Also, you need to choose the article type 'Observations', rather than 'Research Article' right now. Please kindly address these issues.

When submitting the revised version of your paper, please provide (1) point-by-point responses to the issues raised by the reviewers as file type "Response to Reviewers," not in your cover letter, and (2) a PDF file that indicates the changes from the original submission (by highlighting or underlining the changes) as file type "Marked Up Manuscript - For Review Only". Please use this link to submit your revised manuscript. Detailed instructions on submitting your revised paper are below.

Link Not Available

Sincerely,

Takamasa Ueno

Reviewer comments:

Preparing Revision Guidelines

For complete guidelines on revision requirements, please see the journal Submission and Review Process requirements at <https://journals.asm.org/journal/Spectrum/submission-review-process>. **Submissions of a paper that does not conform to**

Microbiology Spectrum guidelines will delay acceptance of your manuscript. "

Please return the manuscript within 60 days; if you cannot complete the modification within this time period, please contact me. If you do not wish to modify the manuscript and prefer to submit it to another journal, please notify me of your decision immediately so that the manuscript may be formally withdrawn from consideration by Microbiology Spectrum.

Responses to the comments of Editor

【Comment】

Thank you for resubmitting the revised version. I think that the revised version well addressed the comments of the reviewer and the scientific evaluation has been well done. However, Observation category requires the guideline '1,200 words with a maximum of 2 figures/tables and 25 references'. Also, you need to choose the article type 'Observations', rather than 'Research Article' right now. Please kindly address these issues.

【Response】

Thank you for your peer review. We appreciate your comments. We have revised the manuscript as an “Observation” style.

April 27, 2023

Dr. Yosuke Hirotsu
Yamanashi Central Hospital
Genome Analysis Center
1-1-1 Fujimi
Kofu, Yamanashi
Japan

Re: Spectrum01316-23R2 (Antibody response to the BA.5 bivalent vaccine shot: a two-year follow-up study following initial COVID-19 mRNA vaccination)

Dear Dr. Yosuke Hirotsu:

Your manuscript has been accepted, and I am forwarding it to the ASM Journals Department for publication. You will be notified when your proofs are ready to be viewed.

Sincerely,

Takamasa Ueno
Editor, Microbiology Spectrum
